

**Comment on "Thermal infrared observations of a western United States biomass burning**
**aerosol plume" by Sorenson et al. (2024)**
Michael D. Fromm[1]
[1]Remote Sensing Division, Naval Research Laboratory, Washington, DC, USA
*Correspondence to*: Michael D. Fromm (michael.d.fromm4.civ@us.navy.mil)
**Abstract.** Sorenson et al. (2024) studied fresh smoke plumes from the proximal Dixie fire in
northern California and concluded that the smoke cooled the air and Earth surface below the
smoke by shielding of incoming solar radiation. The so-attributed cooling was immediate,
sudden and on par with diurnal temperature variations. This comment takes issue with their
conclusions, reasoning, and method. By examining the same case and others, it is shown that the
observed cooling within the smoke plume is caused by plume particulates sufficiently large to
intercept and thereby alter upwelling thermal infrared radiation. The evidence presented is the
same satellite and radar data employed by Sorenson et al., but expanded with temporal
animations. A key element of the new analysis is the demonstration of smoke-associated cooling
at nighttime, a circumstance decoupled from the solar-shielding explanation. The refutation of
the proposed solar-shield-cooling in fresh smokes is an essential refinement of the constraints on
the radiative cause-effect in such conditions.

## 1. Introduction


Sorenson et al. (2024) (Hereafter S24) have claimed observational evidence of a direct Earth-
surface cooling effect approaching 25 K by a fresh, dense biomass-burning smoke plume. They
primarily attribute the cooling to "plume-induced surface insolation reduction," a shielding of
incoming solar radiation by optically dense smoke. They clarify, defend, and elaborate on their
thesis in replies to community and reviewer comments (Sorenson, 2024a, b, c, d).

S24's analysis targets the Dixie fire in northern California, between 20-23 July 2021. Their
specific focus is on the Dixie-fire plume in close proximity to the flaming source, i.e. at distances
less than ~100 km. S24's central data item for determining this large—and sudden—cooling
under smoke comes from polar-orbiting and geostationary satellite-based broadband visible
reflectance and window infrared (IR) brightness temperature (BT) measurements. S24 mention
two additional potential causes of apparent smoke-plume surface cooling as inferred by
depressed window BT: 1. large-particle exhaust such as pyrometeors (McCarthy et al. 2019) or
pyrocumulus hydrometeors, and 2. absorption by gaseous emissions such as $H_2O$. Yet their





analysis leads them to largely dismiss these causes and settle on the visible solar shielding
explanation. The main implication of S24's work is that there is a longwave component to smoke
radiative forcing. Their association of window IR cooling with optically thick smoke prompts
S24 to suggest that "brightness temperature at the thermal IR channels may also be used as
another indirect measurement of AOD when aerosol optical depth is over the detection limit of
the traditional aerosol retrieval methods."
Given the uncertainty with respect to the cause of window IR BT depressions in some near-
source wildfire plumes which appear from space to be nothing more than optically thick plain
smoke (suggested by their monochromatic grayness or tan true color), S24 raise the valid
question regarding the particulate composition within. This puzzle is addressed herein with a
refutation of S24's conclusion of visible shielding of solar radiation. Herein, contradictory
evidence is presented specific to the Dixie fire during the timeframe of S24's analysis. Additional
Dixie-fire dates and other fire events are also presented.
The topic brought to light by S24 and further developed herein stands as an important science
challenge: full understanding of the physical nature of near-source wildfire particulate emissions.
Quantifying the wildfire smoke source term is a quest of measurement campaigns such as
NASA's upcoming INjected Smoke and PYRocumulonimbus Experiment (INSPYRE)
(**https://espo.nasa.gov/inspyre**). It is essential to have an accurate satellite-data framework for
evaluating suborbital remote and in situ measurements of freshly emitted dense smoke.
It is necessary at this point to clarify some terminology central to S24 and this commentary.
"Skin temperature" will refer to Earth-surface temperature. Skin temperature is approximately
what weather satellite broadband window IR BT represents in clear-sky conditions albeit with a
minor correction for water vapor absorption (Schmit et al., 2017). "Surface air temperature,"
routinely measured in situ at 2 m height, is a second metric central to S24. Although the two
quantities are obviously physically related, it is essential to distinguish them. For example,
Schmit et al., (2017) cautioned that geostationary operational environmental satellite (GOES)
window BT is not necessarily representative of 2 m surface air temperature.
In the following sections, this comment addresses and contests specific aspects of S24's analysis,
presents a discussion, and offers concluding remarks.

**2. Multispectral Smoke Reflectance**

In section 3.1, S24 introduced a multispectral reflectance image analysis in pursuit of evidence
consistent with coarse-giant size particles responsible for the depressed window IR BT within the



Dixie smoke plume. They presented a snapshot of the Dixie plume in the early afternoon of 22
July (13:10 local time, 21:10 UTC), afforded by Aqua MODIS. See their Figure 1. This analysis
entailed a focus on mapped MODIS true-color reflectance in combination with images of
reflectance at 1.6 and 2.1 µm (S24 Figure 1), in concert with argumentation about aerosol
microphysics and expectations for the wavelength dependence of top-of-atmosphere (TOA)
reflectance. S24 presented the image snapshot view with a statistical analysis of Dixie-plume
reflectance and window IR depression in comparison to that of nearby smoke-free clear-sky
conditions. They concluded that the lack of a visual and statistical signal of enhanced reflectance
at 2.1 µm sufficed toward their determination that coarse or larger particles were not driving the
plume's window IR BT depressions.

This comment finds S24's analysis to be incomplete and misleading. For one thing, a single
snapshot such as S24's Figure 1 may not adequately represent an evolving wildfire/plume
dynamic. Moreover, the early afternoon setting precedes the typical diurnal maximum in wildfire
energy and extreme pyroconvection (Zhang et al., 2012; Fromm et al., 2010). Hence, Aqua
MODIS sampling is insufficient for drawing conclusions about a changing emissions scenario as
portrayed in S24's central Figure 2 and 6), or in general.

This comment presents an analysis of GOES 16 2.2 µm reflectance imagery surrounding the
Aqua MODIS sample in S24's Figure 1. Enhanced shortwave infrared (SWIR) reflectance is
demonstrated in Fig. 1, images of 0.64 and 2.2 µm reflectance, and 10.3 µm BT on 22 July at
21:10 UTC (Aqua time) and two hours later, 23:10 UTC. The evolution of the visible and SWIR
reflectance enhancements is clarified with animations between 19:00 UTC (~2 h pre-MODIS)
and 23:50 UTC. See Movie S01 and S02 in the Supplement. The snapshots and especially the
animation show the spread of visible smoke from the DIxie Fire past the Nevada and Oregon
border, also portrayed by S24's Figure 1. Within the visible plume, window BT depressions
consistent with S24's analysis are evident (Fig. 1c,f). The addition of the 2.2 µm reflectance
snapshots and animation reveal that as the afternoon progresses, plume signals develop and
advance within the visible-channel plume footprint, almost to the state border. It appears from
the animation that Aqua time approximately represents the apparent onset of SWIR reflectance
enhancements. Qualitatively, these animation sequences can be viewed as showing thickening of
visible optical depth, occasional pyrocumulus, and temporal increase in smoke particle size near
the fire source as inferred from the lag in the onset of SWIR reflectance. Similarly, one can infer
a decrease in particle size toward the downwind extent of the plume where the SWIR signal
degrades relative to the visible reflectance.






### 3. S24 Radar Analysis of Plume


In S24's Section 3.1 they present a "final test of the potential impacts of pyrometeors and
hydrometeors on the observed TIR cooling." Here they use NEXRAD radar from two sites,
Beale Air Force Base (KBBX) and Reno (KRGX). They acknowledge the presence of plume
particles sufficiently large and/or concentrated to create widespread radar echoes from the fire to
far downwind (S24 Figure 3j), yet they "conclude that pyrometeors and hydrometeors are not the
primary cause of the thermal infrared (TIR) cooling signal." This comment takes issue with their
analysis and interpretation, addressing two aspects of the radar analysis.

First, S24's important Figure 2 and surrounding discussion focus on three points close to the
Dixie plume source, two within the plume's influence (orange and green spots in their Figure 2)
and one outside (blue). Note that the orange and green points are roughly 32 and 44 km
downstream of the fire source, respectively. The window IR BT depressions of the plume points
relative to the outside point (S24 Figure 2g) are central evidence for their insolation-shielding
argument as opposed to ash/cloud effect on window IR BT. Except for a brief, strong BT dip at
the orange point—attributed to a pyrocumulus cloud—the entirety of the strong, sudden, and
multi-hour cooling at the orange and green points is considered most likely to be the result of
insolation shielding. However, when one compares S24's Figure 2 and 3, it is obvious that plume
echoes completely cover the orange and green spots. The radar data are later than S24's brief
pyrocumulus by more than an hour yet echoes between ~2-6 km altitude cover the orange and
green points as well as a much wider area.

S24 dismiss the elevated, downwind-edge echoes in their Figure 3 k,l as having "next to no
reflectivity." The distance from their Figure 3J-l downwind spot to the upwind echo edge is ~92
km; indeed a considerable spread of plume particles from the fire source. Moreover, the echoes
are ~125 km from the KRGX radar. At least two radar-centric factors are in play that determine
the areal sensitivity to the Dixie plume. One is beam-broadening, which will limit both KBBX
and KRGX sensitivity to the far downrange plume features. In the case of KBBX, noted by S24
to be situated in a valley (67 m ASL), its orientation with respect to the Dixie fire is in the
direction of mountain peaks between ~1500 and 2000 m, so that beam blocking obscures low-
elevation views of the far downrange plume. Given that the Dixie-fire plume's window IR BT
depressions extend far downwind from the source (S24 Figure 3c) and beyond KBBX and
KRGX traceability, it is reasonable then to conclude that Dixie plume particles persist far from
the source beyond where cloud-particle echoes are "next to no reflectivity." It must also be noted
that the GOES SWIR reflectance (S24 Figure 3b) enhancement has spread far northeast of the
Dixie fire and beyond the radar detection range. See the Supplement, Movie S03 and S24's
Figure 3b.






A more comprehensive visualization of the Dixie fire cloud is offered in the Supplement, Movie
S04, three snapshots from which are shown in Fig. 2. The animation is for 20 July (local time),
from early afternoon to post sunset. The data portrayal is 2 km constant altitude plan position
indicator (CAPPI). The background layer is GOES 17 3.9 µm Dixie fire hotspots in red shades.
Dot markers, from southwest to northeast, are S24's Figure 2 orange and green spots, and S24's
distant-echo spot (Their Figure 3j-l), respectively. The animation shows Dixie-plume echoes
spreading to all three points on two occasions, the first in late afternoon and the second spanning
beyond sunset (~03:40 UTC). The snapshots in Fig. 2a,b capture the pyrocumulus cloud
encroaching on S24's orange point and later the green point, corresponding roughly to the initial,
sudden window IR BT dips in S24 Figure 2. Figure 2c illustrates a continuous swath of echoes
from near the Dixie fire to spots more than 90 km downwind.

Hence cloud-size particulates are independently confirmed in a pattern matching the evolving
satellite visible and window IR features near and far downwind of the Dixie fire. Given that
weather-radar particle-size sensitivity is orders of magnitude weaker than that for thermal IR
radiometer sensors, it is not surprising that these radar illustrations represent a subset of the
entire Dixie smoke cloud.

**4. Night Cross-check**

Section 3.4 of S24 is a "cross-check" of their conclusion that smoke shielding of insolation
causes significant window IR cooling. Their cross check is an examination of a nighttime
satellite image of Dixie fire smoke fortuitously illuminated by moonlight (S24 Figure 7b). The
evidently widespread, thick smoke is unaccompanied by a discernible window IR BT depression
(their Figure 7e), unlike the afternoon conditions the day before and after (their Figure 7a,c,d,f).
They considered this cross-check as determinative of insolation-shielding as the prime factor of
the observed daytime window IR cooling under smoke.

This comment refutes this methodology and conclusion. In this challenge, the Dixie fire plume
on the two dates studied by S24 is exploited, as well as an additional date in the Dixie fire's life:
5 August 2021. The challenge herein also introduces a different wildfire event, that of the
northwest USA wildfire outbreak in September 2020 (Abatzoglou et al., 2021; Mass et al., 2021).
The analysis centers on post-sunset animations of GOES window BT. Animations of window IR
BT are vastly preferable to single snapshots considering expectations of a weak IR signal
imposed on a topographically variable surface. The Dixie-fire dates are 21 July, 23 July, and 5
August 2021. Each run from 03:40–07:00 UTC. They are provided in the Supplement (Movie
S05, S06, and S07). Snapshot images roughly 1 h post sunset are shown in Fig. 3. Dixie fire





hotspots are plainly evident as the cluster of "hot" (white) pixels. Emissions emanating from this
hotspot cluster stand out in each animation. Evidence of the window IR plume signature is the
apparent northeastward (21 and 23 July) and northward (5 August) flow against a static
background representing clear-sky conditions. On all three nights the Dixie plume is
straightforwardly evident. On 21 July, the IR plume is weak relative to that on 23 July and 5
August, but it is still discernable as far away from the source as northwestern Nevada and
southern Oregon. With the visual training afforded by the animations, the stand-alone snapshots
are seen to be sufficient proof of the smoke in window IR on 23 July and 5 August (Fig. 3b-c).
However, this is not the case for 21 July (Figure 3a); the snapshot's plume-related cool-BT
features are neither strong nor widespread enough to stand out against the topography-generated
BT mottling. This may be the case for the 23 July situation S24 exhibited in their Figure 7b,e. It
is also possible that by the time of the nighttime imagery in S24's Figure 7 that the overall
particle size within the Dixie smoke had declined to the point of being transparent in the
longwave window. Regardless, the GOES animation Movie S06 in the Supplement makes clear a
nighttime signature of smoke preceding the S24 sampling.
This comment draws attention to the 5 August animation (Movie S07 in the Supplement), which
captures a second active fire in far northern California. This is the Antelope fire, which started on
1 August (https://www.fire.ca.gov/incidents/2021). The animation reveals even subtle window
IR BT depressions flowing north from the fire. Hence both Dixie and Antelope stand as evidence
for particle-driven "cool" top-of-atmosphere emissions.

A fourth example of nighttime window IR BT depression in smoke is presented in the
Supplement, Movie S08. The situation is on 9 September 2020, a few days after widespread fires
had ignited from western Washington State through Oregon (Abatzoglou et al., 2021; Mass et al.,
2021), adding to an already active fire landscape in northern and central California. Movie S08
in the Supplement shows window IR BT plumes emanating from at least 7 hot fires in California
and Oregon. Although not shown, nighttime visible imagery (akin to S24's image in Figure 7b)
for 9 September 2020 confirms widespread smoke from these fires. Strikingly evident is the
advance of the window IR plume signature flowing well off the Oregon coast over the Pacific
Ocean, far from the flaming sources. This example provides incontrovertible proof of a
particulate-based driver of window IR "cooling" as opposed to the insolation-shielding
explanation of S24.

The four examples above of nighttime window IR BT depression linked to wildfire sources
proves that the solar-shielding explanation of S24 is inadequate to explain all such observed
conditions, day or night.






## 5. Surface radiative response


S24 examined daytime conditions on 22 July 2021 to "test the impact of the smoke plumes on
surface conditions." On this day, 2m air temperature was recorded at two stations representing
near-fire dense smoke and distant faint smoke. Dense smoke was present over the near-fire site
(ASOS station O05) from sunrise onward (not shown). S24's Figure 6b reveals that the surface
temperatures at both stations were equal at sunrise but gradually diverged throughout the day.
This is predictable for a smoke-shrouded site as compared to one in nearly clear-sky condition
throughout the day. Utilizing the ASOS data (S24;
https://mesonet.agron.iastate.edu/request/download.phtml?network=AWOS, last access: 19
February 2025), the AAT-O05 temperature difference at is calculated at all mutual reporting
times between 13:55 and 23:55 UTC, and then the average hourly change thereof. The result is
1.29 K/h. The maximum AAT-O05 surface-temperature difference, +9.78 K at 23:55 UTC,
occurs after 9 h of gradual relative cooling under the dense smoke.

The ASOS station O05 2 m air temperature is compared with GOES 16 "clean" window BT in
Fig. 4. Between 13:55 and 23:59 UTC 22 July, the GOES pixel matched with O05's coordinates
(40.282°N, 121.241°W) is plotted in red; the ASOS value in blue. For approximately 3 h in
morning sunlight both GOES and ASOS temperatures increase while the GOES BT exceeds the
2m ASOS temperature. Thereafter the BT flattens then diminishes while the 2m temperature
increases until peaking at 23 UTC., when the ASOS-GOES difference peaks at 16 K.
Considering the relation between ASOS AAT and O05 surface temperature as opposed to the
ASOS O05-GOES BT relation, it is apparent that GOES BT is not a proxy for skin temperature
throughout the day, and certainly decoupled from 2m air temperature. The question then arises as
to what is causing the window IR BT to diverge from skin-temperatures at this location and
time?

That question is addressed with NEXRAD data, as S24 did for 20 July. The focus is on ASOS
O05. Figure 5 is a map of KBBX reflectivity at 14:00 UTC. The radar data are displayed on the 3
km CAPPI cross-section. The underlying layer in Fig. 5 is GOES 16 3.9 um BT, highlighting the
Dixie Fire hot spots in bright red. Station O05 is ~26 km from the Dixie fire as measured from
the station to the upwind edge of the radar echoes. Echoes form a swath originating over the
Dixie fire and extending north northeast beyond O05. An animation of the 3 km CAPPI
reflectivity from 13:30-23:59 UTC (Supplement, Movie S09) reveals that Dixie fire particulate
emissions are continuous while spreading to and beyond O05 regularly. In the afternoon hours
(when the GOES window BT declines from peak values) reflectivity values increase appreciably
while the detectible smoke cloud extends far beyond O05. Accompanying Movie S09 is Movie
S10, GOES 16 clean window IR BT animated between 13:30 and 23:59 UTC 22 July. The



domain is broadened to include the distant ASOS station AAT, marked along with O05. The
fixed, narrow BT color scale allows discernment of the Dixie-fire IR-sensible emission onset and
spread. To the eye, the IR signal of the Dixie plume begins between 16-17 UTC, local morning.
The BT plume intensifies throughout the day and extends beyond the California/Nevada border
by late afternoon. Considering the radar animation, it is apparent that the window BT depression
within the Dixie plume is attributable to particulates sufficiently large to intercept terrestrial
longwave emissions.

**6. Discussion**

S24 cite appropriate literature on other case studies showing the relationship between optically
dense wildfire smoke plumes and surface cooling. Some of these earlier works (e.g. Westphal
and Toon, 1991 and Robock et al., 1991) were in service of providing observational evidence in
support of Nuclear Winter theory (e.g. Robock et al., 2007). All of the cited literature involved
cases of aged, mesoscale or synoptic-scale smoke plumes that persisted over the study regions
for days or longer. The S24 scenario of nascent smoke in close proximity to the source fire is
quite different than the cited cases. S24 raise the valid question as to the determinant of the
sensible window BT depressions observed in some examples of supposed "dry" smoke
exhausted by nearby energetic fires. S24's examples from California's Dixie fire in July 2021 are
a recurring phenomenon, as acknowledged in their work and in the presentation herein.

S24 showed evidence of sudden window IR BT reduction exceeding 10°C, e.g. their Figure 2g.
Focusing solely on the green spot's BT evolution (avoiding the orange point's pyrocumulus
signal), there is a BT reduction of ~19°C within one hour. Visible reflectance jumped
simultaneously. I.e. sky conditions over the green point went from clear to dense smoke between
~22:00 and 23:00 UTC. Simply considering Earth-surface radiative inertia, it would seem
physically implausible for skin temperature to drop so suddenly and greatly solely as a
consequence of smoke visible AOD ramping up as depicted in S24 Figure 2g.

Optically dense wildfire smoke plumes are intuitively known for their tan/brown hue in true
color satellite imagery and gray in monochromatic visible imagery owing to their absorptive
carbonaceous content. It is natural to interpret these shades as "dry," i.e. plain smoke and thereby
infer a certain microphysical/radiative character. However, S24 draw attention to the counter-
intuitive nature of certain dense smokes that appear to embody some cloudlike IR characteristics.
Such conditions are still poorly understood in terms of the smoke-plume composition. However,
there is a subset of plain-looking smoke plumes for which there is strong evidence of comingled
cloud material.



It has been established that pyrocumulonimbus-injected plumes can counter-intuitively manifest
as both opaque aerosol and $H_2O$ cloud. The term "smoke cloud" is introduced for this situation.
The smoke cloud has a strong, cold window IR BT signature and the visible appearance of plain
smoke, i.e brownish or gray in true color or monochromatic imagery, respectively (Fromm and
Servranckx, 2003; Fromm et al., 2005, 2008). It is therefore conceivable for this peculiar
condition to manifest within certain fresh, vigorous smoke plumes, especially ones pockmarked
with capping pyrocumulus turrets (i.e. whitish cloud appearance). In this case the "cloud"
material within the smoky looking plume is likely composed of pyrometeors (McCarthy et al.,
2019), cloud particles, or a blend. This is the case for the Dixie fire plumes analyzed by S24 and
others presented herein.

S24 attribute window IR cooling to fine-mode smoke abruptly "shutting off" insolation
(Sorenson, 2024c), characterizing the "drastic" effect with BT-depression values between 10 and
25 K. This range is on par with the observed clear-sky diurnal BT range shown in their Figure 2.
They do not provide an adequate physical explanation for the rapid and strong window IR BT
decline they attribute to smoke shadowing. While it is reasonable to expect negative forcing on
skin and near-surface air temperature by virtue of plume (or cloud) insolation shielding, sudden
cooling on par with or exceeding the full diurnal temperature/BT cycle is irreconcilable with
empirical Earth-surface response to solar radiation. Put simply, sudden onset of cloudiness or a
dense plume does not typically drive 20+ K skin- or air-temperature reductions

It is important to recognize that weather radar data are inherently subject to range-dependent
sensitivity degradation due to beam broadening. Convolving this effect with radar
wavelength/particle-size sensitivity, it is expected that certain portions of a smoke cloud will fall
below detectability thresholds in comparison to the inherently near uniform satellite-sampling in
combination with window IR wavelength sensitivity to sub-millimeter particles. In short, the
absence of radar echoes does not imply no particles. Smoke clouds are preferentially discerned
by satellite in the window IR while near-range active radar remote sensing confirms the "cloud"
qualification.

The most concise distillation of S24's conclusions is given in their abstract: "…clear signals in
water vapor and TIR channels suggest that both co-transported water vapor injected to the
middle to upper troposphere and surface cooling by the reduction of surface radiation by the
plume are more significant, with the surface cooling effect of smoke aloft being the most
dominant." The water-vapor-injection argument is not supportable. S24 present no observational
support for pyrogenic injection to the middle or upper troposphere. Indeed, their messaging
explicitly excludes the scenario of such deep pyroconvection except for an isolated pyrocumulus
cloud on 20 July (local time) briefly perturbing GOES window IR BT. Implicit in S24's
argumentation is that the overall smoke plume's effect on IR BT is that there is little to no
evidence of cloud-size particles therein. This argument limits the injection height to levels below



cloud condensation levels, i.e., to the lower troposphere. Hence it is illogical to imply mid- to
upper-tropospheric transport of the smoke plume by virtue of involving such upper-level water
vapor signals.

**7. Conclusions**

This comment was inspired by the publication of Sorenson et al., (2024), who addressed the
admittedly curious condition of apparently dry smoke plumes freshly emitted from wildfires
accompanied by thermal IR absorption. S24's proposition is that opaque, supposedly dry smokes
can dramatically cool the Earth surface by effectively shutting off insolation.

Their case studies and two new ones were presented wherein an alternate, intuitive explanation is
manifest. It was revealed that in fact particulate matter within the fresh plumes is responsible for
the IR absorption in every case. The most convincing evidence is thermal IR absorption in the
smoke at night from the same fire (Dixie) S24 studied. S24's position is that their scenario would
only apply to daytime.

In addition to S24's July 2021 Dixie fire episodes, another approximately two weeks later was
introduced. Once again deploying GOES thermal IR image animations, it was shown that
nighttime IR smoke emissions from Dixie as well as a second fire (Antelope) were present.
Finally, a remarkable nighttime scenario from September 2020 in the northwest USA was
presented. On this occasion, IR emissions spewed from at least 7 fires in Oregon and California,
emissions that were evident even over the distant Pacific Ocean.

In addition to deploying GOES image data, NEXRAD reflectivity data were examined, as in
S24. It was shown that particulate-generated-reflectivity enhancements were present where S24
reported solar-shielding-attributed cooling at two weather stations. Arguments were presented for
the value of radar data for smoke-plume detection and also the various limitations of these data
relative to satellite broadband imagery.

The GOES analysis also consisted of visible and shortwave IR reflectance views, as in S24.
Animations from midday to evening improved the plume-detection capability and revealed
multispectral reflectance enhancements indicative of sufficiently large smoke particles,
countering S24's assessment that SWIR reflectance was absent and by extension, no indication
of coarse-mode particles.



The subject of smoke composition in fresh, dense plumes is ripe for future exploration.
McCarthy et al. (2019)'s revelation of pyrometeors is a starting point for pondering and
diagnosing the particulate composition and microphysics of smoke plumes that represent the
source condition for a host of science questions. Future measurement campaigns such as NASA's
INjected Smoke and PYRocumulonimbus Experiment (INSPYRE)
(**https://espo.nasa.gov/inspyre**) have science aims including characterizing the smoke-plume
source term. It is essential to have an accurate foundational construct for the physical properties
of freshly emitted dense smoke. This reinterpretation of S24's conclusions is thereby offered.





## Figures and Captions

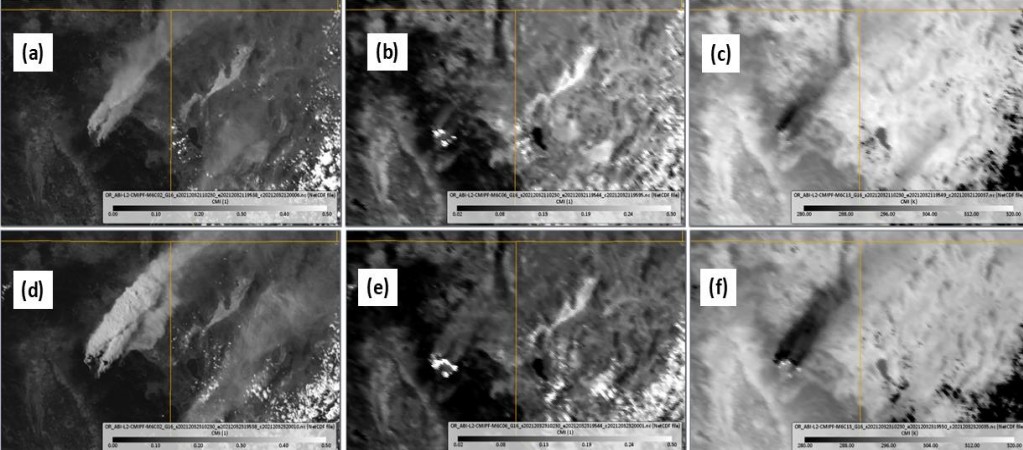

**Figure 1.** GOES East multi-spectral images focused on northern California's Dixie fire and plume, 22 July 2021. Top row, 21:10 UTC; bottom row, 23:10 UTC. Left column, 0.64 µm "red"channel reflectance. Middle column, 2.2 µm channel shortwave-IR reflectance. Right column, 10.3 µm "clean" window brightness temperature (BT). See the various color-bar legends for the reflectance (unitless) and BT (K) ranges.

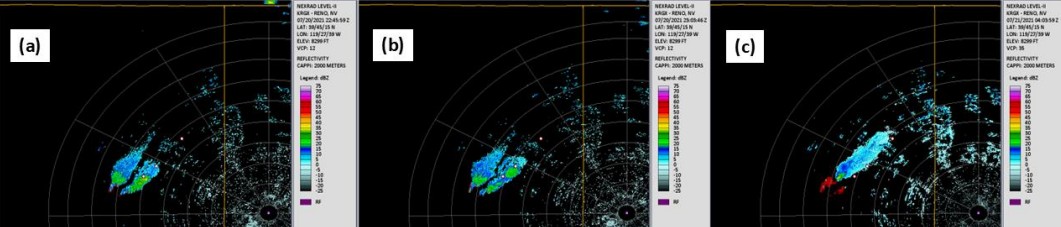

**Figure 2.** Reno, Nevada NEXRAD (site ID KRGX, purple dot mark) 2 km constant altitude plan position indicator (CAPPI) image for 22 July 2021. The two southwestward red-white dots are at S24's Figure 2 orange- and green-dot locations. The marker toward the northeast represents S24's Figure 3j plume-reflectivity edge. a) 22:46 UTC. b) 23:04 UTC. c) 04:05 UTC 21 July (post sunset).





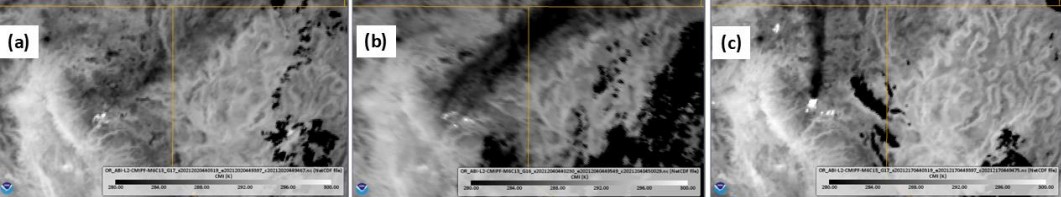

**Figure 3.** Dixie-fire area post-sunset (04:40 UTC) GOES "clean" window (10.3 μm) IR BT images. a) 21 July, GOES West. b) 23 July, GOES East. c) 5 August, GOES West.

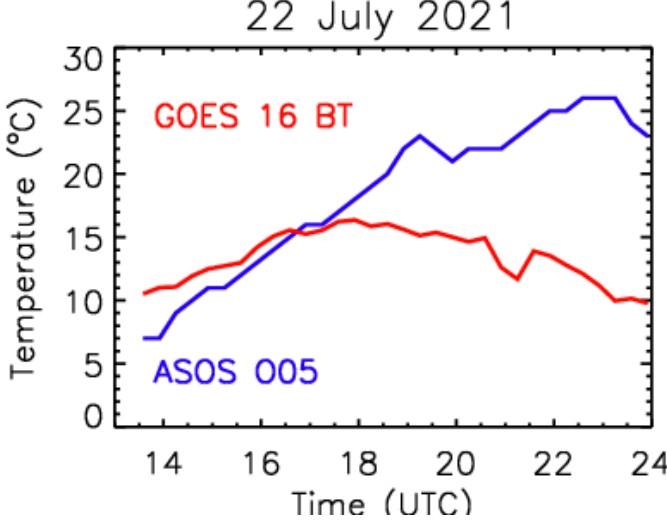

**Figure 4.** ASOS station O05 temperature and GOES West window BT time series, daytime 22 July. The ASOS data (blue line) are the same as in S24's Figure 6. Red line, BT for the GOES pixel closest to O05 location and reporting time (See text for coordinates).



412

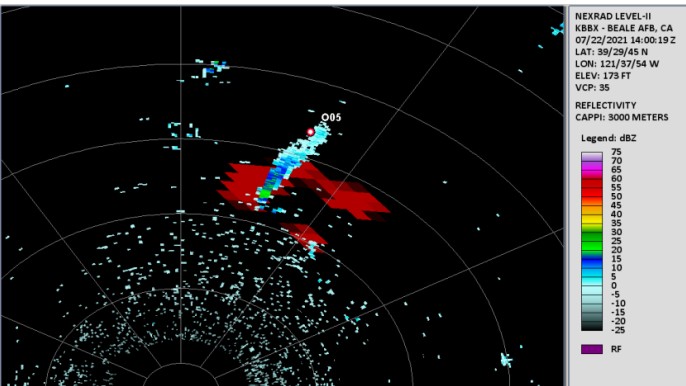

413

**Figure 5.** Beale Air Force Base (site ID KBBX) NEXRAD 3 km CAPPI reflectivity, 22 July,
14:00 UTC. Radar location, purple dot. Range rings start at 10 km, thereafter 25 km spacing.
Background layer: GOES 3.9 μm BT; red pixels signify the Dixie fire hot spots. ASOS station
O05 is marked with a dot and annotated.

*Acknowledgements:* MDF is grateful to Gerald Nedoluha, who provided helpful suggestions for
the manuscript.

**Data Availability**

ASOS data were retrieved from

https://mesonet.agron.iastate.edu/request/download.phtml.

GOES image data are part of the NOAA Big Data initiative. https://registry.opendata.aws/noaa-
goes/. They were accessed via the NOAA Weather and Climate Toolkit:
https://www.ncdc.noaa.gov/wct/

*Competing interests*: The author has no competing interests.

*Financial support*: Support for this work was entirely from US Naval Research Lab base
program.

433



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
