# Peer review of "Comment on "Thermal infrared observations of a western United States biomass burning"

_EGUsphere, 2025_

## Author Comment (AC3)

Reply to Sorenson et al. (2025)'s response to Fromm (2025) Comment.

Sorenson et al. (2025) will be abbreviated as "S25."

My ACP Comment will be referred to as "F25".

**General Statement**

When S24 was still in the ACP discussion phase I posted a Community Comment because of my doubts about S24's insolation-shielding explanation for the extensive, persistent TIR BT reduction in the fresh Dixie Fire smoke plume. I suggested the use of NEXRAD data in hopes that the examination of these would persuade S24 of a more plausible explanation, that being large particles perturbing the absorption and emission in the TIR. Unfortunately, S24 persisted in their overall attribution of observed TIR cooling in the fresh smokes to insolation shadowing. Thus, I felt compelled to submit the Comment (F25). Again unfortunately, S25 demonstrate the continued need to maintain the argument for solar shielding in spite of the evidence I provided that TIR cooling continues seamlessly in darkness, a condition antithetical to the solar shielding position. S25's continued de-amplification of the radar evidence and misleading description of all of the radar echoes as "immediately downwind" of the fire is, in my assessment, unsupportable. Consequently, I hold that S24 and S25 fail to make the case for insolation shielding as the cause for observed satellite-based TIR BT depressions in fresh smoke. Below I provide a set of major concerns with S25, followed by a list of technical issues.

**Major Concerns**

In Section 3.3, S25 focus on 5 August 2021, one of three days in the life of the Dixie fire to which F25 drew attention (the others being S24's two July Dixie case studies.). They inaccurately connect the F25 analysis timeframe to a pyroCb. F25's nighttime emissions with IR BT cooling preceded S25's pyroCb by ~14 hours. I acknowledge though that S25 did concur with F25's finding as a consequence of their pre-sunrise 5 August analysis. Their concurrence should logically redound to F25's three other nighttime scenarios (21 and 23 July 2021, and 9 September 2020); the only explanation for nighttime smokes exhibiting thermal IR cooling is that which F25 offered. Moreover, accepting this explanation also redounds to the daytime hours adjacent to these night Dixie-plume episodes, when thermal-IR-perturbed smoke seamlessly spans the day/night transition. I.e., why does one need to invoke an additional, novel explanation for IR-perturbed smoke when the IR signature occurs at all times of the day?

In Section 3.3 S25 continue the 5 August study in afternoon hours when extreme pyroconvection is witnessed. Curiously, S25 equivocate on the sole role of particle absorption/emission in the midst of acknowledged deep, expansive pyrocloud material. As with the prior evening (and those of 20 and 22 July), IR-perturbed smoke continues seamlessly past sunset, on this occasion flowing far into Nevada and Oregon (not shown but verifiable via the same means both author groups have deployed). Moreover, GOES 2.2 micron reflectance enhancements follow the IR plume right up to sunset. In summary, there is no compelling requirement to seek an explanation other than the particle IR absorption/emission phenomenon, given all the evidence in S24, S25, and F25.

S25 continue S24's invocation of Bragg Scattering as a potentially spurious creator of radar echoes. This phenomenon can be dismissed given that it would, at the very least, have to be active on all of the presented Dixie Fire episodes, encompassing the entire echoing feature, and all times illustrated. Otherwise, the robustness of the radar-echo observations stands as proof of plume particulates large enough to perturb thermal IR emissions as observed. Arguing, as S24 and S25 do, that Bragg Scattering could be a contributor to echo intensity leaves unstated the implication that the other contributor is large particles. Finally, the radar-echo features offered in the various figures of S24, S25, and F25 all expand tens of kilometers away from the active Dixie flaming to locations that are not subject to the potential turbulence within which Bragg Scattering is hypothesized.

With regard to the NEXRAD data and S25's presentation, I found no acknowledgement or challenge of F25's caveats about weather radar sensitivity limits. In short, it must be restated that <no echoes> does not necessarily mean <no particles> (F25). This realization is clear when examining all of the sets of KMAX, KBBX, and KRGX PPI visualizations in S25. Considering any of these triplets, one sees that the Dixie plume has a shape/footprint unique to each radar. This is a straightforward demonstration of the relative advantages/disadvantages of each radar's view of a single plume at the same instant in time. One cannot conclude that a lack of echoes in one radar's view means no particles when/where another radar reveals their presence. Radar data such as those presented in S24, S25, and F25 must be accepted as a blunt tool relative to satellite-based broadband IR sensitivity to hydrometeors, chaff, debris, ash, etc. Even so, active radar remote sensing represents an indisputable source of observations of large smoky particles, not only at the fire source but as far as 90 km downstream of the Dixie Fire (F25).

S25 introduce a new case (29 June 2015 over the upper Midwest USA) they contend shows depressed window BT within a smoke plume with no discernable NIR visible reflectance. Given that this is smoke not attached to an active fire, nor attributed to a particular fire's source, it is somewhat incomparable to the premise of S24. As in S24, their evidence is founded on a single MODIS snapshot. F25 pointed out the weakness of individual snapshots rather than animations. This is especially relevant to situations wherein signals such as NIR reflectance and window BT perturbations are faint, as is the case here. It is conceded that a weak BT depression in the smoke is evident. But it is not conceded that the NIR reflectance is nil, a condition central to S25's argument. Regarding the apparent BT-depression swath, it is evident to me that the coordinates of the Grand Forks AERONET station place these observations at the eastern fringe of the BT-depressed feature at Aqua MODIS time (I used NASA's Worldview, https://worldview.earthdata.nasa.gov/, to assess this.). Hence the association of the sizable fine-mode fraction with the smoke within the BT-depression is questionable.

Given S25's agreement with F25 in the case of 5 August 2021, their argument that the daytime scenes of the Dixie Fire plume involve some blend of insolation shutting off and large-particle-imposed TIR BT reduction is difficult to accept. If the relatively cool smoke plumes that flow continuously from the Dixie Fire to 100+ km distant don't have a single determinant, the difficulty of proving an additional factor is enormous. In my assessment, S24 and S25 did not accomplish that. The explanations given did not have sufficient rigor. For example, S25 used an example of post sunset on 21 July (Their Figure 4) to posit that an elevated plume at terminator time (that was not visible in the satellite imagery) might cast a shadow sufficient to cause the intense TIR cooling observed at that time. Even if there was such a plume element, the insolation at the terminator is too weak to have imposed any such effect.

On nine instances within S25 the term "immediately downwind" is used to characterize Dixie Fire radar echoes with respect to the flaming source. This term is not defined; it seems to be used to create the impression that the entirety of radar-echo features is all near enough to the flaming and associated turbulence to invite the invocation of Bragg Scattering and/or placing limits on the spread of large particles to the fire's direct influence. In actuality, the preponderance of the coherent echo features to which they refer extend well beyond the fire. See for example, S25's Figure 3e, f, wherein "immediately downwind" echoes stretch completely across a California county, 50+ km from the source. This is a demonstrable misapplication of the potential for Bragg Scattering to be causal.

S25 took issue with F25's Figure 4 and attendant analysis, which suggested that ASOS 2-m temperature was decoupled from supposed skin temperature as represented by collocated GOES TIR BT. S25 proposed complicating factors such as GOES pixel size, parallax-related geo-registration error, and lake proximity, and that F25 did not account for advection by prevailing winds. First, S24 did not account for temperature advection either; they took ASOS station temperatures at face value. Secondly, it is evident from the plume motion (F25) that low-level winds were southwesterly, meaning that advection from the lake toward the GOES pixel (and ASOS site) was not a factor. Thirdly, parallax error is inconsequential if the emitting surface is the ground. Finally, I redid my GOES analysis with the adjacent pixel west of that reported in F25. The BT pattern (not shown but available upon request) was like that in F25's Figure 4. On average throughout the period illustrated on Figure 4, the BT of the adjacent pixel to the west (which has no overlap with the lake) was ~1.5 K colder than F25's. Considering the overall plume spread on this day, shown by both S24 and F25, the decoupling of the ASOS air temperature and thermal emissions from this location is further supported.

S25's solar eclipse case study is stimulating while being largely inapplicable to F25's Comment. Equating a total obscuration of the Sun by a solid object (the moon) to a diffuse, albeit dense smoke plume is not compelling. If S25's aim is to lend credence to S24's attribution of sudden 10-25 K TIR cooling to sudden shadowing by dense smoke, it is not convincing. As a reminder, the locations at which S24 reported the steep, sudden drop in thermal IR BT (S24's Figure 2g) were covered by radar echoes (S24, F25).

F25 acknowledged that obscuration of the Sun could lead to some Earth-surface cooling. An example like the total eclipse (S25) is a dramatic illustration of this effect. And of course, TIR cooling of the Earth surface happens every day as insolation diminishes. But the evidence accumulated within S24, F25 and S25 clearly shows another determinant of in-smoke TIR cooling in hours-old plumes, meaning that support for S24's hypothesis has not yet been established.

In Section 3.2, last paragraph of page 10, S25 stated, "F25 disagreed with the VIIRS nighttime analysis presented in S24, claiming that GOES-16 data showed TIR cooling signals after sunset in the very early UTC hours on 23 July 2021" and followed up with arguments I could not understand. First, F25 did not simply "claim" TIR BT cooling after sunset; it was illustrated in Movie S06. Whether or not that cooling persisted until S24's VIIRS image time is irrelevant. Unless S25 dispute the veracity and implications of Movie S06 (and F25's derived Figure 3b), it stands as a refutation

of the insolation-shielding argument. Note that S24 and S25 defend the sudden (and on occasion stark) onset of TIR BT cooling as a reasonable artifact of smoke shadowing. In that scenario, the moment that the sun sets the TIR BT cooling should vanish. This is not what is observed, according to F25.

S24's response to Reviewer #2: "If the smoke particles themselves were causing any nighttime TIR cooling signal, we would expect any surface features to become obscured by a uniform region of decreased temperatures;…" This is precisely what F25 showed on four different nights. S24's reviewer reply continued: "…thus, since clear surface features are visible in the nighttime imagery, the smoke particles themselves are not causing TIR cooling at night." That is justifiable. But as F25 showed, S24's single snapshot at night on 23 July doesn't represent all night conditions, in particular it didn't even represent the entire night of 23 July.

**Technical Issues**

Lake outlines are not shown in the relevant S25 figure panels. It is impossible to know what portion is obscured without outlines or before/after imagery (or both).

Eagle Lake is 44 km from the fire on 5 August. I.e., it is not "immediately" downwind of the fire.

Section 2.1, Page 6, top lines: F25 did not use the term "bubble" nor focus on what S25 characterized as a "small area of locally enhanced TIR cooling" post sunset on 20 July. Rather, F25 drew attention to a protracted TIR-BT-depression feature flowing from the Dixie fire over several post-sunset hours, evident over and beyond Nevada (F25's Movie S05).